# Direct Amination of Nitroquinoline Derivatives via Nucleophilic Displacement of Aromatic Hydrogen

**DOI:** 10.3390/molecules26071857

**Published:** 2021-03-25

**Authors:** Jakub Wantulok, Daniel Swoboda, Jacek E. Nycz, Maria Książek, Joachim Kusz, Jan Grzegorz Małecki, Vladimír Kubíček

**Affiliations:** 1Faculty of Science and Technology, Institute of Chemistry, University of Silesia in Katowice, ul. Szkolna 9, PL-40007 Katowice, Poland; jakub.wantulok1@gmail.com (J.W.); daniel.swoboda@us.edu.pl (D.S.); jan.malecki@us.edu.pl (J.G.M.); 2Faculty of Science and Technology, Institute of Physics, Univeristy of Silesia in Katowice, 75 Pułku Piechoty 1a, 41-500 Chorzów, Poland; maria.ksiazek@us.edu.pl (M.K.); joachim.kusz@us.edu.pl (J.K.); 3Faculty of Pharmacy in Hradec Králové, Charles University Prague, Akademika Heyrovského 1203, 500 05 Hradec Králové, Czech Republic; kubicek@faf.cuni.cz

**Keywords:** amination, vicarious nucleophilic substitution of hydrogen, nucleophilic aromatic substitution (S_N_Ar), nitration, heterocyclic, Skraup, Meldrum’s acid

## Abstract

The vicarious nucleophilic substitution of hydrogen (VNS) reaction in electron-deficient nitroquinolines was studied. Properties of all new products have been characterized by several techniques: MS, HRMS, FTIR, GC-MS, electronic absorption spectroscopy, and multinuclear NMR. The structures of 4-chloro-8-nitroquinoline, 8-(*tert*-butyl)-2-methyl-5-nitroquinoline, 9-(8-nitroquinolin-7-yl)-9*H*-carbazole and (*Z*)-7-(9*H*-carbazol-9-yl)-8-(hydroxyimino)quinolin-5(8*H*)-one were determined by single-crystal X-ray diffraction measurements. The 9-(8-nitroquinolin-7-yl)-9*H*-carbazole and (*Z*)-7-(9*H*-carbazol-9-yl)-8-(hydroxyimino)quinolin-5(8*H*)-one illustrate the nitro/nitroso conversion within VNS reaction. Additionally, 9-(8-isopropyl-2-((8-isopropyl-2-methyl-5-nitroquinolin-6-yl)methyl)-5-nitrosoquinolin-6-yl)-9*H*-carbazole is presented as a double VNS product. It is postulated that the potassium counterion interacts with the oxygen on the nitro group, which could influence nucleophile attack in that way.

## 1. Introduction

The worldwide annual production of quinoline derivatives is more than 2000 tonnes, of which 8-hydroxyquinoline makes up the main part [1]. The amino group is the essential building block of many molecules found in nature, e.g., peptides and proteins. Introducing this moiety into molecular structures is a crucial process, no matter the scale of the reaction and the purpose of the application. To explore the structure-function relationship of biologically active quinoline compounds, such as their coordination metal ability, the study of amination of nitroquinoline derivatives was carried out in our laboratory. This research focuses on the vicarious nucleophilic substitution of hydrogen (VNS). Recently, the amination reactions of 4,7-dichloro-1,10-phenanthrolines using bulky 9*H*-carbazole, 10*H*-phenothiazine, and pyrrolidine nucleophiles were reported by our group [2]. The targeted compounds obtained were exclusively 4,7-diamino-1,10-phenanthrolines with yields up to 96% [2]. Two fully characterized products of a similar chemical transformation, the oxidative nucleophilic substitution of hydrogen (ONSH), were presented (Figure 1) [2].

Both reactions, VNS and ONSH, proceed via the initial addition of nucleophile to the nitroaromatics and heteroaromatics to form a sigma-complex type intermediate. Then spontaneous oxidation of intermediate sigma-complex by oxidizing agents produces final products. The advantage of these methods is the aromatic functionalization with no need for halogenated materials or expensive metal catalysts. The work of Prof. Mąkosza and co-workers inspired us to perform our investigations [3].

## 2. Results and Discussion

In the current study, we present the amination procedures of selected nitroquinoline derivatives.

### 2.1. Synthesis of Nitroquinoline Derivatives

In order to synthesize nitroquinoline derivatives as the precursors of the target aminoquinolines, two types of chemical transformations were applied. One was based on the classical Skraup-Doebner-Miller reaction and a three-step cyclocondensation of 2,2-dimethyl-1,3-dioxane-4,6-dione (Meldrum’s acid), trimethyl orthoformate, and 2-nitroaniline (molecule **4b**). The synthetic routes and their structures are presented in Scheme 1. In the second route, we chose the direct nitration of selected 8-(alkyl)-2-methylquinolines (molecules **3c** and **3d**), which was a one-step method to introduce carbon-nitrogen bond and requires higher electron density in benzene (or phenol) rings as described earlier [4]. The nitration of hydroxyquinoline derivatives occurs at low temperatures ca. 5 °C due to the hydroxyl group’s strong activating effect. In comparison, the syntheses of nitroquinolines **4c** and **4d** required a temperature of at least 70 °C and took more than 16 h to complete. In all cases, reactions were selective and occurred with high yields, up to 94% (Scheme 1).

The presented nitroquinoline derivatives **4** are compounds that have easily formed crystals (Figure 2 and Figure 3).

### 2.2. X-ray Studies of Nitroquinolines

The molecules **4b** and **4c** crystallized in orthorhombic *P*bc21 and monoclinic *P*21/c space groups, respectively. Their molecular structures are displayed as ORTEP representations in Figure 2. The compounds **4b** and **4c** are planar. As one can see from Figure 3, the compound **4c** has two independent molecules in the asymmetric unit. The structure of the presented molecule **4c** is stabilized by π-π stacking interactions, which occur between quinoline rings (Figure 3). The centroid-centroid distances vary from 3.668–3.734 Å, and the shift distances are between 1.233–1.464 Å.

Additional bond lengths and angles and crystallographic refinement details can be found in Appendix A from Appendix A.

### 2.3. Amination of Nitroquinoline Derivatives via Vicarious Nucleophilic Substitution of Hydrogen

Crystalline nitroquinoline derivatives **4a**, **4b**, and **4d** with the presence of hindered and non-hindered hydrogens in *ortho* and/or *para* positions were chosen in our studies. The readily available aromatic hydrogen located in *ortho* and/or *para* position to the nitro group is the main requirement for the vicarious nucleophilic substitution from the starting material. The nitro group activates an aromatic ring to nucleophilic attack. Nucleophilic addition to carbon atoms of the nitroaromatics and heteroaromatics is a fast and reversible process [3], during which negatively charged intermediates are created, i.e., the Meisenheimer complex (Scheme 2), with the stabilization of substituents via the delocalization of charge. Therefore, electron-withdrawing (EW)-type substituents, especially the nitro group, are needed. The nucleophile’s counterion, such as potassium cation, is attracted by the nitro group’s negatively charged oxygen atoms (Scheme 2). The resulting adducts to restore the aromaticity have to lose hydride anions (Scheme 2). This process could be realized by oxidation by external oxidants. N. J. Lawrence et al. suggests that the nitro group also plays a role in the oxidation processes [5]. It is essential to mention that VNS reaction may compete with the aromatic nucleophilic substitution of halogen S_N_Ar. The substitution of the nitro substituent is also possible. According to Mąkosza et al., the VNS often proceeds much faster than substitutions, as mentioned above [3,5,6]. The VNS reaction is usually highly colored, which has diagnostic values [3]. However, some exceptions could be seen in literature, such as 4-fluoro-5-nitropyridine, which participates exclusively in the aromatic nucleophilic substitution of halogen S_N_Ar [7].

In our studies, as a nucleophile, 9*H*-carbazole was chosen to compare our recent results from the amination reactions of 4,7-dichloro-1,10-phenanthrolines [2]. 9*H*-Carbazole has a special rigid planar structure and is a valuable building block for the synthesis of many products like drugs or innovative materials.

The commercially available quinolone **4a** and 9*H*-carbazole were initially selected to carry out the reaction to ascertain the mechanism and optimize conditions. According to previously reported data, the reaction of molecule **4a** was carried out with an excess (1.5 equiv.) of potassium 9*H*-carbazol-9-ide in THF solvent, which quickly turned red, at reflux temperature according to previously reported data [2,8]. Crystalline 9-(8-nitroquinolin-7-yl)-9*H*-carbazole (**5a**) and (*Z*)-7-(9*H*-carbazol-9-yl)-8-(hydroxyimino)quinolin-5(8*H*)-one (**5b**) were isolated with low yield (Scheme 2, and Figure 4 and Figure 5). At first sight, it is somewhat surprising that the vicarious nucleophilic substitution via nucleophilic displacement of aromatic hydrogen of molecule **4a** led to two products, both with 9*H*-carbazolyl group located exclusively at the C7 position. The regiochemistry of this type of substitution is strongly affected by the size of the nucleophile. Because of the bulky nucleophile used, it was expected that a newly formed C-N bond will be formed rather in C5 position than C7. This finding suggests the potassium counterion assists the interaction with the nitro group. A nucleophile’s attack generates a Meisenheimer adduct (σ^H^-adduct) followed by a loss of hydrogen to restore the aromaticity (Scheme 2). The molecule **5a** possesses at C7 position a newly-attached 9*H*-carbazolyl substituent in ortho location to nitro group (Scheme 2). The molecule **5b’** contains a hydroxyl group at C5 and nitroso substituent at C8 position. The conversion of nitro to nitroso group is known in literature [3,9]. In our research to generate potassium carbazol-9-ide in THF, potassium *tert*-butoxide and 9*H*-carbazole were chosen. The in-situ-generated *tert*-BuOH serves as a proton source (Scheme 2; step 1). The nitro group at the C8 position in molecule **5a** or Meisenheimer adduct in protic media could be transformed into nitroso by protonation and the subsequent elimination of a water. The origin of compound **5b’** requires both the transformation of the nitro group into nitroso and the presence of hydroxyl substituent (or carbonyl) at C5 position. One explanation of the origin of the **5b’** could be the displacement of the potassium hydroxide from the Meisenheimer adduct in protic media (Scheme 2; step 2). The potassium hydroxide could attack Meisenheimer’s nitrosophenyl ring at C5 position and form a hydroxyl group in nitroso adduct, followed by oxidation by oxygen from the air or nitro substituent to a carbonyl group (Scheme 2; step 3). T. N. Gurova et al. showed the influence of substituents and medium on the tautomeric equilibrium between nitrosophenols and quinone oximes [10]. Another similar phenomenon is reported by I. R. Baxendale et al. [11]. We report a similar type of phenomenon between the nitroso adduct **5b’** and molecule **5b** (Scheme 2).

We already mentioned that VNS reaction might compete with the aromatic nucleophilic substitution of halogen S_N_Ar. In order to verify this thesis in the subsequent reaction, quinolone **4b** with chloride atom located in activated C4 position was selected. The reaction was carried out with the same condition as previously reported. Our results showed that the direct nucleophilic displacement of an aromatic hydrogen reaction proceeds together with expected aromatic nucleophilic substitution of halogen S_N_Ar. The reaction mixture was very complicated. However, we were able to identify the product similar to 9,9’-(8-nitroquinoline-4,7-diyl)bis(9*H*-carbazole) (5c) as a result of VNS and S_N_Ar subsequent substitutions. GC-MS’s identified product is with tr = 27.3 min, (EI) M^+^ = 508 (100%), whose structure is similar to molecule **5c** and has an exact mass is 504 (Appendix A from Appendix A, Scheme 3).

The N-C_Aryl_ bond formation via VNS reaction in 2-methylquinoline derivatives such as molecule **4d** is more complicated due to the acidic protons located on the methyl group C2 position [12]. Their presence in the quinoline constitution implicates a competition reaction between VNS N-C_Aryl_ bond formation and a base attack at acidic protons located on the methyl group in C2 position. To synthesize the targeted 9-(8-isopropyl-2-methyl-5-nitrosoquinolin-6-yl)-9*H*-carbazole (VNS product; Scheme 3), a reaction was carried out where molecule **4d** was treated with excess (1.5 equiv.) potassium 9*H*-carbazol-9-ide in THF solvent at reflux temperature under the same reaction conditions as the above example. Molecule **4d** was chosen due to the presence of an easier diagnostic *iso*-propyl group. Unexpectedly a product of double VNS substitution 9-(8-isopropyl-2-((8-isopropyl-2-methyl-5-nitroquinolin-6-yl)methyl)-5-nitrosoquinolin-6-yl)-9*H*-carbazole (**5d**) was obtained. The formation of nitroso adduct observed was possible by elimination of a water molecule, which is similar to the aforementioned mechanism of origin of **5b’** (Scheme 2). The origin of molecule **5d** can be explained by potassiation with potassium *tert*-butoxide followed by an intramolecular transfer of the 8-*iso*-propyl-2-methyl-5-nitroquinoline, which afforded molecule **5d** in 44% yield (Scheme 3). N. J. Lawrence et al. reported VNS reaction product as an intermediate react with various electrophiles [5].

As one can see from Figure 4, the compound **5a** has three independent molecules in the asymmetric unit and molecule **5b** two in Figure 5. Selected bond lengths (Å) and angles (°) for **5a** are collected in Appendix A from Appendix A.

A characteristic feature in ^1^H-NMR spectra of molecule **5b** with hydroxyimino group that was acquired in dimethyl sulfoxide (DMSO) solvent was the pronounced downfield shift of OH proton signal (17.41 ppm), pointing to a strong intramolecular N···HO-N hydrogen bond. The same OH proton signal was moved to 17.76 ppm in CDCl_3_ solvent. The explanation of this phenomenon is the ability to form a pseudo-ring by molecule **5b**, stabilized by intramolecular hydrogen bond in the DMSO or CDCl_3_ solvent (see Scheme 2, Figure 5). The structure of compound **5b** confirmed by X-ray structural analysis showed strong intramolecular N···HO-N hydrogen bonds, which are listed in Table 1. Selected bond lengths (Å) and angles (°) for **5b** are collected in Appendix A from Appendix A.

The molecules **5a** and **5b** were crystallized in triclinic *P*-1 and orthorhombic *P*na21 space groups, respectively. Their molecular structures are displayed as ORTEP representations in Figure 4 and Figure 5 in planar orientation. In Figure 4, compound **5a** has three, and compound **5b** has two independent molecules in the asymmetric unit (Figure 5). The structure of molecule **5b** is stabilized by strong intramolecular N···HO-N hydrogen bonds (Table 1, Figure 5) and π-π stacking interactions, which occurs between quinoline rings (Figure 6). The centroid–centroid distances vary from 3.586–3.642 Å, and the shift distances are between 0.206–1.050 Å (Figure 6). 

Compounds **5a** and **5b** are composed of an 8-nitroquinoline core in a planar orientation and peripheral amino substituent. The 9*H*-carbazol-9-yl substituent is strongly twisted in relation to the quinoline rings with a larger angle (86.43°, 84.98°, 67.96°) for **5a** and (~54.73°) for **5b** between quinoline and 9*H*-carbazol-9-yl substituent planes in the position C7. It is worth noting that bond lengths between the carbon atom at C7 position in the quinoline moiety and nitrogen atom in 9*H*-carbazol-9-yl substituent are ~1.42 Å and are comparable to the values of 1.42 Å in 4,7-di(9*H*-carbazol-9-yl)-9-oxo-9,10-dihydro-1,10-phenanthroline-5-carbonitrile [2].

### 2.4. UV-Vis Studies for Molecule ***5b***

Our studies were performed by varying pH and solvent to examine their effect on the absorbance of molecule **5b** (Figure 7). Compound **5b** shows the ability to form a pseudo-ring, stabilized by an intramolecular hydrogen bond in solution, shown on ^1^H-NMR spectra as a signal from OH group (in CDCl_3_ solution 17.76 ppm). The phenomenon of forming a pseudo-ring, which was stabilized by intramolecular hydrogen bond, was also observed before by our group [13]. Additionally, the tautomerism between nitrosophenol **5b’** and quinine oxime **5b** forms presented in Scheme 2 could have an influence on absorption spectra. In Figure 7 we present the graph confirming the presence of a pure anionic form of nitrosophenol **5b**’ in 1 M KOH solution for studied molecule **5b** (black line). The similar tendency was observed for **5b** in dimethylformamide (DMF) solvent (red line). The λ_max_ value of compound **5b** in methanol in comparison with KOH and DMF solutions showed bathochromic shift (red shift), which could be explained by the presence of hydrogen bonds between quinine oxime **5b** and methanol molecules (green dots). As expected, the increase of the number of hydrogen bonds between solute and more acidic CHCl_3_ leads to an increase in the wavelength of the absorption maxima of compound **5b**, which shifts bathochromically (blue intermittent line). The neutral DMSO and acetonitrile (ACN) solvents showed intermediate results. We observed longer wavelengths (bathochromic shift) with a decrease of dielectric constant in the absorption band. Additionally, the dielectric dependence on the wavelength was presented in Figure 7. 

In the ^1^H-NMR spectra in DMSO and CDCl_3_ polar solvents, we observed only nitroso form **5b**. However, after the addition of a drop of KOD solution of D_2_O to DMSO solvent in the ^1^H-NMR spectrum we did not see the characteristic OH proton signal (17.41 ppm) (Appendix A from Appendix A). The aromatic signals were moved significantly to upfield (smaller δ), which suggests the presence of pure nitrosophenol **5b’**. Additionally, we observed hydrogen-deuterium (H/D) exchange between water-*d*_2_ KOD solution and aromatic protons. The experimental (UV-Vis and NMR studies for molecule **5b**) and computational analysis (Appendix A from Appendix A) revealed that only in KOH and DMF solutions can we expect the presence of nitroso form **5b’**.

The nitroquinoline **4b** with chlorine atom at C4 position was chosen in the next reaction, as the main difference complying with already used **4a**. In this case, two reactions were possible, i.e., direct nucleophilic displacement of aromatic hydrogen via VNS or S_N_Ar of chloride located in activated C4 position. The molecule **4b** was treated with excess (1.5 equiv.) potassium 9*H*-carbazol-9-ide in THF solvent at reflux temperature, similarly to the previous experiment. The reaction was much more complex. The product with only a substituted chlorine atom at C4 position by 9*H*-carbazole suggests a S_N_Ar reaction. On the other hand, a product with only substituted aromatic hydrogen at C7 position by 9*H*-carbazole suggests the occurrence of VNS reaction. Our results showed a complex reaction mixture. The product identified only by GC-MS suggests a structure of molecule **5c**, which proves that both reactions proceed.

### 2.5. Amination of Nitroquinoline Derivatives via Nitro Group Reduction with Stannous Chloride

The obtained nitroquinolines **4** were easily reduced to appropriate aminoquinolines **6** in mild conditions. The synthetic route and structures of the aminoquinoline derivatives **6** are presented in Scheme 4. This method is suitable for various nitroquinolines **4** and tolerate to the presence of many useful functional groups, including hydroxyl, alkyl substituents, and halogen atoms. The reaction offers a simple and fast transformation, with yields of up to 86%.

The ^1^H-NMR solution spectra of aminoquinolines **6** showed distinctive H-1 signals from NH_2_ group proton with chemical shift *ca*. 4.9 ppm (C8-NH_2_) or ca. 4.1 ppm (C5-NH_2_). For all presented compounds **4**, **5**, and **6**, distinctive H-1 and C-13 signals come from alkyl groups.

The analysis of the trends in ^1^H and ^13^C chemical shifts revealed that the aminoquinolines **6** molecules had significantly increased shielding effect (upfield effect, smaller δ) in comparison with nitroquinolines 4 in CDCl_3_ solvent.

### 2.6. X-ray Studies

Crystals of compounds **4b**, **4c**, **5a**, and **5b** were mounted in turn on a Gemini A Ultra Oxford Diffraction and SuperNova automatic diffractometer (Agilent Technologies, Santa Clara, CA, USA) equipped with a CCD detector used for data collection. X-ray intensity data were collected with graphite monochromated or with a micro-focused MoKα radiation at room temperature, with ω scan mode. Details concerning crystal data and refinement are gathered in Table 1. Lorentz, polarization, and empirical absorption correction using spherical harmonics implemented in SCALE3 ABSPACK scaling algorithm were applied [14]. The structures were solved by a direct method and subsequently completed by a difference Fourier recycling. All the non-hydrogen atoms were refined anisotropically using full-matrix, least-squares techniques. The Olex2 [15], SHELXS and SHELXL [16] programs were used for all the calculations. Atomic scattering factors were incorporated in the computer programs. Details concerning crystal data and refinement are gathered in Table 2.

## 3. Materials and Methods

### 3.1. Materials

All experiments were carried out in an atmosphere of dry argon, and flasks were flame dried. Solvents were dried by usual methods (diphenyl ether, diethyl ether, and THF over benzophenone ketyl, CHCl_3_, and CH_2_Cl_2_ over P_4_O_10_, hexane over sodium-potassium alloy) and distilled. Chromatographic purification was carried out on silica gel 60 (0.15–0.3 mm, Macherey-Nagel GmbH & Co. KG, Düren, Germany). Sodium hydride (dry, 95%), potassium tert-butoxide, trimethyl orthoformate, 2-iso-propylaniline, 2-(tert-butyl)aniline, 2-nitroaniline, phosphoryl chloride, 9*H*-carbazole, Meldrum’s acid, acrolein, and crotonaldehyde were purchased from Sigma-Aldrich (Poznań, Poland), and were used without further purification.

### 3.2. Instrumentation

NMR spectra were obtained with Avance 400 and 500 spectrometers (Bruker, Billerica, MA, USA) operating at 500.2 or 400.2 MHz (^1^H) and 125.8 or 100.6 MHz (^13^C) at 21 °C. Chemical shifts referenced to ext. TMS (^1^H, ^13^C) and ext. DSS (^1^H, ^13^C), or using the residual CHCl_3_ signal (δ_H_ 7.26 ppm) and CDCl_3_ (δ_C_ 77.1 ppm) as internal references for ^1^H and ^13^C-NMR, respectively. Coupling constants are given in Hz. For GC-MS, a 7890A gas chromatograph (Agilent Technologies, Wilmington, DE, USA) was equipped with a MS (70 eV) 5975 EI/CI MSD, and a 7693 autosampler with an Agilent HP-5MS capillary column (30 µm × 250 μm × 0.25 μm) press. 127.5 kPa, total flow 19 mL/min, col. flow 2 mL/min, split-7:1, temp. prog. (70 °C-hold 0.5 min, 70–290 °C/25 °C/min., 290 °C-hold 6 min) was used. The LCMS-IT-TOF analysis was performed on an Agilent 1200 Series binary LC system coupled to a micrOTOF-Q system mass spectrometer (Bruker Daltonics, Brema, Germany). High-resolution mass spectrometry (HRMS) measurements were performed using a Synapt G2-Si mass spectrometer (Waters, New Castle, DE, USA) equipped with an ESI source and quadrupole-time-of-flight mass analyser. To ensure accurate mass measurements, data were collected in centroid mode and mass was corrected during acquisition using leucine enkephalin solution as an external reference (Lock-SprayTM, Waters, New Castle, DE, USA). The measurement results were processed using the MassLynx 4.1 software (Waters, Milford, MA, USA) incorporated within the instrument. A Nicolet iS50 FTIR spectrometer was used for recording spectra in the IR range 4000–400 cm^−1^. FTIR spectra were recorded on a Perkin Elmer (Schwerzenbach, Switzerland) spectrophotometer in the spectral range 4000–450 cm^−1^ with the samples in the form of KBr pellets. Elementary analysis was performed using Vario EL III apparatus (Elementar, Langenselbold, Germany). Melting points were determined on MPA100 OptiMelt melting point apparatus (Stanford Research Systems, Sunnyvale, CA, USA) and are uncorrected.

### 3.3. Synthesis of 8-(tert-Butyl)-2-methylquinoline (***3c***), 8-(iso-Propyl)-2-methylquinoline (***3d***) and 8-Nitroquinoline (***4a***)

The synthesis of quinolines **3c**, **3d,** and **4a** followed our procedure described in the literature [17].

Toluene (50 mL) and crotonaldehyde (2.6 mL, 2.2 g, 31.4 mmol) were added to a solution of 2-(*tert*-butyl)aniline (2.3 g, 15.7 mmol) in aqueous 6 M HCl (200 mL) and were heated under reflux for 16 h. The mixture was allowed to cool to room temperature. The aqueous layer was separated and neutralized with an aqueous solution of K_2_CO_3_. After extraction with CH_2_Cl_2_ (3 × 50 mL), the organic layer was separated and dried over MgSO_4_, and then was filtered and distilled bp 110–115 °C/ 3 mmHg. The liquid was purified by crystallization from hexane to afford white crystals:

*8-(tert-Butyl)-2-methylquinoline* (**3c**) [18] 2.4 g (12.2 mmol, 78%); mp = 55.1–56.3 °C; ^1^H-NMR (CDCl_3_; 400.2 MHz) δ = 1.68 (s, 9H, C(CH_3_)_3_), 2.72 (s, 3H, CH_3_), 7.21 (d,^3^*J*_H,H_ = 8.4 Hz, 1H, aromatic), 7.36 (t,^3^*J*_H,H_ = 7.7 Hz, 1H, aromatic), 7.60 (m, 2H, aromatic), 7.98 (d,^3^*J*_H,H_ = 8.4 Hz, 1H, aromatic); ^13^C{^1^H}-NMR (CDCl_3_; 100.6 MHz) δ = 25.5, 31.0, 36.5, 120.6, 125.0, 125.8, 126.3, 127.2, 136.4, 146.9, 147.5, 155.6.

Toluene (50 mL) and crotonaldehyde (2.6 mL, 2.2 g, 31.4 mmol) were added to a solution of 2-*iso*-propylaniline (2.1 g, 15.7 mmol) in aqueous 6 M HCl (200 mL) and were heated under reflux for 16 h. The mixture was allowed to cool down to room temperature. The aqueous layer was separated and neutralized with aqueous solution of K_2_CO_3_. After extraction with CH_2_Cl_2_ (3 × 50 mL), the organic layer was separated and dried over MgSO_4_, and then was filtered and distilled bp 100–110 °C/ 3 mmHg. The liquid mixture was dissolved in concentrated 36% HCl (100 mL) at 5 °C, and ZnCl_2_ (2.7 g, 20.0 mmol) was added with vigorous stirring for 1 h. The precipitate was filtered, washed with cold 3 M aq. HCl and dried in air. The solid was washed with *i*PrOH and dried. The received white solid was added to 10% ammonia solution and extraction with Et_2_O (3 × 50 mL); the organic layer was separated and dried over MgSO_4_ to afford greenish oil:

*8-(iso-Propyl)-2-methylquinoline* (**3d**) [19] 2.1 g (11.6 mmol, 74%); bp 100–110 °C/3 mmHg; ^1^H-NMR (CDCl_3_; 400.2 MHz) δ = 1.37 (d,^3^*J*_H,H_ = 7.0 Hz, 6H, CH(CH_3_)_2_), 2.73 (s, 3H, CH_3_), 4.39 (septet,^3^*J*_H,H_ = 6.9 Hz, 1H, CH), 7.23 (d,^3^*J*_H,H_ = 8.3 Hz, 1H, aromatic), 7.42 (t,^3^*J*_H,H_ = 7.6 Hz, 1H, aromatic), 7.57 (t, ^3^*J*_H,H_ = 7.1 Hz, 2H, aromatic), 7.98 (d,^3^*J*_H,H_ = 8.4 Hz, 1H, aromatic); ^13^C{^1^H}-NMR(CDCl_3_; 125.8 MHz) δ = 23.6, 25.7, 26.9, 121.4, 125.0, 125.2, 125.4, 126.4, 136.3, 145.7, 146.8, 157.5.

Toluene (50 mL) and acrolein (2.1 mL, 1.8 g, 31.4 mmol) were added to a solution of 2-nitroaniline (2.2 g, 15.7 mmol) in aqueous 6 M HCl (200 mL) and were heated under reflux for 16 h. The mixture was allowed to cool to room temperature. The aqueous layer was separated and neutralized with aqueous solution of K_2_CO_3_. After extraction with CH_2_Cl_2_ (3 × 50 mL), the organic layer was separated and dried over MgSO_4_, and then was filtered and evaporated to afford yellowish crystals:

*8-Nitroquinoline* (**4a**) [20] 2.6 g (14.8 mmol, 94%); mp = 90.1–91.3 °C; ^1^H-NMR (CDCl_3_; 400.2 MHz) δ = 7.57 (dd,^3^*J*_H,H_ = 8.4 Hz, ^3^*J*_H,H_ = 4.2 Hz, 1H, aromatic), 7.63 (t,^3^*J*_H,H_ = 7.9 Hz, 1H, aromatic), 8.05 (2d, ^3^*J*_H,H_ = 7.9 Hz, 2H, aromatic), 8.28 (d,^3^*J*_H,H_ = 8.4 Hz, ^4^*J*_H,H_ = 1.4 Hz, 1H, aromatic), 9.08 (dd, ^3^*J*_H,H_ = 4.1 Hz, ^3^*J*_H,H_ = 1.4 Hz, 1H, aromatic); ^13^C{^1^H}-NMR(CDCl_3_; 100.6 MHz) δ = 122.8, 123.8, 125.3, 129.1, 132.1, 136.2, 139.6, 147.5, 152.7.

### 3.4. Synthesis of 4-Chloro-8-nitroquinoline (***4b***)

#### 3.4.1. Step A

Trimethyl orthoformate (406.2 g, 500 mL, 3830.0 mmol) and Meldrum’s acid (21.6 g, 150.0 mmol) were heated to a gentle reflux for 30 min. The resulting greenish solution was cooled to 80 °C and 2-nitroaniline (15.0 g, 108.7 mmol) was added portion wise (exothermic reaction). The resulting mixture was stirred up to reflux for 2 h, and left under room temperature (rt) for 16 h. Subsequently, hexane was added and the solution was cooled to −35 °C where a precipitate formed. The precipitate was filtered off, washed with diethyl ether (4 × 100 mL), and dried to afford a white solid:

*2,2-Dimethyl-5-(((2-nitrophenyl)amino)methylene)-1,3-dioxane-4,6-dione* (**2a**) [21] 28.2 g (96.7 mmol, 89%); mp = 175–178 °C; ^1^H-NMR (CDCl_3_; 400.2 MHz) δ = 1.77 (s, 6H, 2CH_3_), 7.41 (d, ^3^*J*_H,H_ = 7.8 Hz, 1H, aromatic), 7.66 (d, ^3^*J*_H,H_ = 8.3 Hz, 1H, aromatic), 7.79 (t, ^3^*J*_H,H_ = 7.7 Hz, 1H, aromatic), 8.32 (d, ^3^*J*_H,H_ = 7.6 Hz, 1H, aromatic), 8.75 (d, ^3^*J*_H,H_ = 13.6 Hz, 1H, vinyl), 13.02 (d, ^3^*J*_H,H_ = 13.6 Hz, 1H, NH); ^13^C{^1^H}-NMR(CDCl_3_; 100.6 MHz) δ = 27.3, 91.4, 105.5, 118.0, 125.9, 127.0, 134.4, 136.1, 138.1, 151.2, 163.3, 164.2; UV-Vis (methanol; λ (nm) (logε)): 357 (4.59), 308 (4.68), 274 (4.49), 256 (4.40), 216 (4.67); IR (KBr): ν = 3161, 3094, 3004, 1733, 1686, 1604, 1518, 1343, 1265, 1201, 933, 741 cm^−1^.

#### 3.4.2. Step B

Into freshly distillated diphenyl ether (50 mL) at 220 °C was added **2a** (2.92 g, 10.0 mmol) in small portions, resulting in vigorous gas evolution. The resulting orange solution was brought to reflux for 30 min and was then allowed to cool to 50 °C. The hexane (25 mL) was added and a brown solid precipitated was filtered and washed with hexane (2 × 10 mL). The crude product was purified by crystallization from chloroform/hexane mixture to yield solid as follows:

*8-Nitro-4(1H)-quinolinone* (**3a**) [21] 1.6 g (8.4 mmol, 84%); mp = 192–194 °C; ^1^H-NMR (DMSO-*d_6_*; 500.2 MHz) δ = 6.24 (d, ^3^*J*_H,H_ = 7.5 Hz, 1H, aromatic), 7.50 (d, ^3^*J*_H,H_ = 8.0 Hz, 1H, aromatic), 7.97 (d, ^3^*J*_H,H_ = 7.5 Hz, 1H, aromatic), 8.56 (d, ^3^*J*_H,H_ = 7.9 Hz, 1H, aromatic), 8.63 (d, ^3^*J*_H,H_ = 7.9 Hz, 1H, aromatic), 11.86 (s, 1H, NH); ^1^H-NMR (CDCl_3_; 400.2 MHz) δ = 6.23 (dd, ^3^*J*_H,H_ = 7.6 Hz, ^4^*J*_H,H_ = 1.8 Hz, 1H, aromatic), 7.97 (dd, ^3^*J*_H,H_ = 7.6 Hz, 1H, aromatic), 8.54–8.57 (m, 1H, aromatic), 8.58–8.64 (m, 1H, aromatic), 11.86 (bs, 1H, NH); ^13^C{^1^H}-NMR(CDCl_3_; 100.6 MHz) δ = 112.5, 122.2, 128.5, 130.3, 134.9, 135.5, 138.4, 177.1; ^13^C{^1^H}-NMR(DMSO-*d_6_*; 125.8 MHz) δ = 110.5, 121.8, 127.7, 129.7, 133.5, 134.1, 136.5, 141.2, 175.6; UV-Vis (methanol; λ (nm) (logε)): 382 (4.07), 256 (4.30), 220 (4.51), 207 (4.54); IR (KBr): ν = 3331, 3234, 1643, 1605, 1564, 1490, 1314, 1291, 1245, 1188, 1068, 745 cm^−1^.

#### 3.4.3. Step C

Into freshly distillated phosphoryl chloride (82.0 g, 50 mL, 534.8 mmol) under argon, **3a** (0.9 g, 5.0 mmol) was mixed, and the resulting solution was stirred at 90 °C for 4 h. The excess of phosphoryl chloride was slowly evaporated under reduced pressure. The reaction mixture was slowly added to a well-stirred mixture of ice (50 g) in water (100 mL). After stirring for 15 min, the resulting reaction mixture was carefully brought to pH 13–14 by adding NaOH solution (40%). The aqueous layer was extracted with CH_2_Cl_2_ (4 × 10 mL). The combined organic layers were separated and dried over MgSO_4_. Evaporation of the brown-colored solvent afforded **4b** as light tan crystals. Next, the crude products were purified by chromatography on silica gel using methanol/dichloromethane as eluent and finally crystallization from CH_2_Cl_2_ to yield precipitates as follows:

*4-Chloro-8-nitroquinoline* (**4b**) [22] 0.8 g (4.0 mmol, 81%); mp = 120–125 °C; ^1^H-NMR (DMSO-*d_6_*; 400.2 MHz) δ = 7.93 (dd, ^3^*J*_H,H_ = 8.1 Hz, ^3^*J*_H,H_ = 7.9 Hz, 1H, aromatic), 7.98 (d, ^3^*J*_H,H_ = 4.7 Hz, 1H, aromatic), 8.39 (dd, ^3^*J*_H,H_ = 7.5 Hz, ^4^*J*_H,H_ = 0.8 Hz, 1H, aromatic), 8.48 (dd, ^3^*J*_H,H_ = 8.5 Hz, ^4^*J*_H,H_ = 0.9 Hz, 1H, aromatic), 8.98 (d, ^3^*J*_H,H_ = 4.7 Hz, 1H, aromatic); ^13^C{^1^H}-NMR (DMSO-*d_6_*; 100.6 MHz) δ = 123.4, 124.1, 126.3, 127.40, 127.42, 139.3, 141.9, 148.2, 152.5; GC-MS: t_r_ = 7.228 min, (EI) M^+^ = 208 (100%), (M − NO_2_)^+^ = 162 (33%); GC-MS: t_r_ = 7.3 min, (EI) M^+^ = 208.1 (100%); UV-Vis (methanol; λ (nm) (logε)): 316 (4.23), 302 (4.30), 283 (4.47), 216 (5.02); IR (KBr): ν = 3047, 1958, 1847, 1535, 1484, 1358, 880, 865, 750, 717 cm^−1^; CCDC (The Cambridge Crystallographic Data Centre) 1967406.

### 3.5. Synthesis of 8-(Alkyl)-2-methyl-5-nitroquinolines ***4c*** and ***4d***

8-(Alkyl)-2-methylquinoline **3c** or **3d** (7.5 mmol) was dissolved in a mixture of concentrated H_2_SO_4_ and HNO_3_ (4.5 and 10.5 mL, respectively) at 5 °C. After stirring for 1 h at room temperature, no evolution of gas was observed, so the reaction mixture was heated up to 70 °C and stirred overnight. After this time, the reaction mixture was poured down to a beaker containing 25 g of ice and 25 mL of water and the precipitated solid was filtered off, washed with 10 mL of cold water, and dried on air, giving:

*8-(tert-Butyl)-2-methyl-5-nitroquinoline* (**4c**) as a yellow solid. 1.1 g (4.6 mmol, 61%); mp = 86.1–87.3 °C; ^1^H-NMR (CDCl_3_; 500.2 MHz) δ = 1.69 (s, 9H, C(CH_3_)_3_), 2.77 (s, 3H, CH_3_), 7.45 (d,^3^*J*_H,H_ = 8.9 Hz, 1H, aromatic), 7.70 (d,^3^*J*_H,H_ = 8.3 Hz, 1H, aromatic), 8.17 (d,^3^*J*_H,H_ = 8.3 Hz, 1H, aromatic), 8.85 (d,^3^*J*_H,H_ = 8.9 Hz, 1H, aromatic); ^13^C{^1^H}-NMR(CDCl_3_; 125.8 MHz) δ = 25.4, 31.1, 37.7, 119.9, 123.2, 123.6, 124.4, 132.0, 144.6, 146.8, 155.9, 157.2; UV-Vis (methanol; λ (nm) (logε)): 317 (4.02), 281 (3.94), 258 (4.00), 224 (4.66), 202 (4.61); IR (KBr): ν = 2959, 1909, 1609, 1517, 1500, 1333, 827, 801 cm^−1^; Anal. Calcd for C_14_H_16_N_2_O_2_: C, 68.83; H, 6.60; N, 11.47; O, 13.10 Found: C, 69.00; H, 6.66; N, 11.33; CCDC (The Cambridge Crystallographic Data Centre) 2048040.

*8-(iso-Propyl)-2-methyl-5-nitroquinoline* (**4d**) as a beige solid. 1.3 g (5.7 mmol, 76%); mp = 59.1–60.3 °C; ^1^H-NMR (CDCl_3_; 500.2 MHz) δ = 1.38 (d,^3^*J*_H,H_ = 6.9 Hz, 6H, CH(CH_3_)_2_), 2.79 (s, 3H, CH_3_), 4.47 (septet,^3^*J*_H,H_ = 6.9 Hz, 1H, CH), 7.50 (d,^3^*J*_H,H_ = 8.9 Hz, 1H, aromatic), 7.64 (d,^3^*J*_H,H_ = 8.1 Hz, 1H, aromatic), 8.28 (d,^3^*J*_H,H_ = 8.1 Hz, 1H, aromatic), 8.92 (d,^3^*J*_H,H_ = 8.9 Hz, 1H, aromatic); ^13^C{^1^H}-NMR(CDCl_3_; 125.8 MHz) δ = 23.3, 25.4, 27.9, 119.4, 123.5, 123.8, 124.4, 132.2, 143.5, 145.3, 155.3, 159.0; UV-Vis (methanol; λ (nm) (logε)): 321 (4.06), 225 (4.67), 202 (4.55); IR (KBr): ν = 2963, 1960, 1895, 1606, 1517, 1449, 1342, 803 cm^−1^; Anal. Calcd for C_13_H_14_N_2_O_2_: C, 67.81; H, 6.13; N, 12.17; O, 13.90 Found: C, 67.94; H, 6.19; N, 12.01.

### 3.6. Syntheses of 9H-Carbazol-9-yl-8-nitroquinolines ***5a*** and ***5b***

These were based on the procedure described in the literature [8]. To the suspension of *tert*-BuOK (1.01 g, 9.04 mmol) in THF (50 mL), 9*H*-carbazole (1.1 g, 6.58 mmol) was added and reagents were stirred under reflux for 30 min under argon. 8-Nitroquinoline **4a**, **4b,** or **4d** (4.52 mmol), respectively was then added to the reaction mixture, which was refluxed overnight. After the evaporation of the solvent to give a solid, water (20 mL) and CH_2_Cl_2_ (100 mL) were added. The organic layer was separated and the aqueous layer was extracted with CH_2_Cl_2_ (4 × 50 mL). The combined organic layers were dried over MgSO_4_. After solvent evaporating, the crude product was purified by column chromatography on silica gel using methanol/dichloromethane as eluent to afford a crude solid, and finally crystallization from a mixture of CH_2_Cl_2_ and hexane to yield solids as follows:

*9-(8-Nitroquinolin-7-yl)-9H-carbazole* (**5a**) 0.11 g (0.3 mmol, 7%); mp = 160–163 °C; ^1^H-NMR (CDCl_3_, 500.2 MHz) δ = 7.20 (dt, ^3^*J*_H,H_ = 8.1 Hz, ^4^*J*_H,H_ = 0.9 Hz, 2H, aromatic), 7.32 (ddd, ^3^*J*_H,H_ =8.1 Hz, ^4^*J*_H,H_ = 7.2 Hz, ^4^*J*_H,H_ = 1.0 Hz, 2H, aromatic), 7.40 (ddd, ^3^*J*_H,H_ =8.3 Hz, ^4^*J*_H,H_ = 7.2 Hz, ^4^*J*_H,H_ = 1.3 Hz, 2H, aromatic), 7.60 (d, ^3^*J*_H,H_ = 8.7 Hz, 1H, aromatic), 7.69 (dd, ^3^*J*_H,H_ = 8.4 Hz, ^4^*J*_H,H_ = 4.3 Hz, 1H, aromatic), 8.12 (ddd, ^3^*J*_H,H_ = 7.8 Hz, ^4^*J*_H,H_ = 1.3 Hz, ^4^*J*_H,H_ = 0.8 Hz, 2H, aromatic), 8.16 (d, ^3^*J*_H,H_ = 8.7 Hz, 1H, aromatic), 8.39 (dd, ^3^*J*_H,H_ = 8.4 Hz, ^4^*J*_H,H_ = 1.7 Hz, 1H, aromatic), 9.14 (dd, ^3^*J*_H,H_ = 4.3 Hz, ^4^*J*_H,H_ = 1.7 Hz, 1H, aromatic); ^13^C{^1^H}-NMR (CDCl_3_; 125.8 MHz) δ = 110.2, 120.6, 121.2, 123.6, 124.3, 126.6, 127.8, 128.8, 130.5, 131.3, 136.1, 140.5, 141.5, 148.0, 153.3; HRMS (IT-TOF): *m*/*z* Calcd for C_21_H_14_N_3_O_2_ (M + H)^+^ = 340.1086, Found 340.1094; UV-Vis (methanol; λ (nm) (logε)): 329 (3.74), 315 (3.74), 287 (4.21), 225 (4.88), 206 (4.69); IR (KBr): ν = 3060, 2925, 1731, 1541, 1450, 1225, 757 cm^−1^; CCDC (The Cambridge Crystallographic Data Centre) 2048038.

*(Z)-7-(9H-Carbazol-9-yl)-8-(hydroxyimino)quinolin-5(8H)-one* (**5b**) 0.24 g (0.7 mmol, 16%); mp_dec._ = 210–211 °C; ^1^H-NMR (CDCl_3_, 500.2 MHz) δ = 6.97 (s, 1H, aromatic), 7.31 (ddd, ^3^*J*_H,H_ = 8.0 Hz, ^4^*J*_H,H_ = 4.7 Hz, ^4^*J*_H,H_ = 3.4 Hz, 2H, aromatic), 7.42 (dd, ^3^*J*_H,H_ = 3.6 Hz, ^4^*J*_H,H_ = 1.0 Hz, 4H, aromatic), 7.84 (dd, ^3^*J*_H,H_ =8.0, ^4^*J*_H,H_ = 4.9, 1H, aromatic), 8.10 (dt, ^3^*J*_H,H_ =7.8, ^4^*J*_H,H_ = 1.0, 2H, aromatic), 8.81 (dd, ^3^*J*_H,H_ =8.0, ^4^*J*_H,H_ = 1.8 Hz, 1H, aromatic), 8.88 (dd, ^3^*J*_H,H_ =4.9 Hz, ^4^*J*_H,H_ = 1.8 Hz, 1H, aromatic), 17.76 (s, 1H, OH); ^1^H-NMR (DMSO-*d_6_*, 500.2 MHz) δ = 7.03 (s, 1H, aromatic), 7.29 (dd, ^3^*J*_H,H_ = 7.4 Hz, 2H, aromatic), 7.43 (dd, ^3^*J*_H,H_ = 7.3 Hz, 2H, aromatic), 7.56 (d, ^3^*J*_H,H_ = 8.2 Hz, 2H, aromatic), 8.03 (dd, ^3^*J*_H,H_ = 8.0 Hz, ^4^*J*_H,H_ = 4.9 Hz, 1H, aromatic), 8.21 (d, ^3^*J*_H,H_ =7.7 Hz, 2H, aromatic), 8.74 (dd, ^3^*J*_H,H_ =8.0 Hz, ^4^*J*_H,H_ = 1.5 Hz, 1H, aromatic), 9.07 (dd, ^3^*J*_H,H_ = 4.9 Hz, ^4^*J*_H,H_ = 1.5 Hz, 1H, aromatic), 17.41 (s, 1H, OH); ^1^HNMR (DMSO-*d_6_*/KOD, 500.2 MHz) δ = 7.19–7.33 (m, ^3^*J*_H,H_ = 7.0 Hz, ^4^*J*_H,H_ = 0.8 Hz, 4H, aromatic), 7.40 (t, ^3^*J*_H,H_ = 7.4 Hz, 2H, aromatic), 7.55 (dd, ^3^*J*_H,H_ = 7.9 Hz, ^4^*J*_H,H_ = 4.4 Hz, 1H, aromatic), 8.23 (d, ^3^*J*_H,H_ =7.6 Hz, 2H, aromatic), 8.66 (dd, ^3^*J*_H,H_ =8.1 Hz, ^4^*J*_H,H_ = 2.0 Hz, 1H, aromatic), 8.94 (bs, 1H, aromatic); ^13^C{^1^H}-NMR (CDCl_3_; 125.8 MHz) δ = 111.2, 120.5, 121.2, 124.6, 126.0, 126.3, 126.4, 126.7, 136.7, 140.2, 141.3, 147.6, 148.1, 150.2, 182.4; MS (IT-TOF): *m*/*z* (relative intensity (rel. int.) (M + H)^+^ = 340.1091 (100%); (M − H_2_O)^+^ = 322.0984 (23%);(M + Na)^+^ = 362.0913 (9%); HRMS (IT-TOF): *m*/*z* Calcd for C_21_H_14_N_3_O_2_ (M + H)^+^ = 340.1086, Found 340.1091; UV-Vis (1 M KOH; λ (nm) (logε)): 406(4.22), 330 (3.83), 318 (3.86), 287 (4.10), 276 (4.13), 234 (4.63); (DMF; λ (nm) (logε)): 430 (3.56), 330 (4.13), 317 (4.18); (DMSO; λ (nm) (logε)): 430 (3.00), 331 (3.77), 320 (3.81), 290 (4.03); (ACN; λ (nm) (logε)): 444 (2.64), 330 (3.63), 317 (3.71), 289 (3.78), 228 (4.10); (methanol; λ (nm) (logε)): 453 (3.22), 328 (4.13), 317 (4.17), 287 (4.37), 228 (4.69); (THF; λ (nm) (logε)): 459(3.18), 330 (4.11), 318 (4.15), 288 (4.34), 239 (4.55)**;** (CDCl_3_; λ (nm) (logε)): 478(3.35), 329 (4.23), 318 (4.24), 289 (4.42); IR (KBr): ν = 3045, 2927, 1649, 1601, 1447, 1298, 750, 719 cm^−1^; CCDC 2048039.

*9-(8-isoPropyl-2-((8-isopropyl-2-methyl-5-nitroquinolin-6-yl)methyl)-5-nitrosoquinolin-6-yl)-9H-carbazole* (**5d**) 0.61 g (1.0 mmol, 44%); mp = 290.1–291.3 °C; ^1^H-NMR (CDCl_3_, 500.2 MHz) δ = 1.48 (d, ^3^*J*_H,H_ = 6.8 Hz, 6H, CH(CH_3_)_2_), 1.56 (d, ^3^*J*_H,H_ = 7.0 Hz, 6H, CH(CH_3_)_2_), 2.80 (s, 3H, CH_3_), 4.24 (septet,^3^*J*_H,H_ = 6.8 Hz, 1H, CH), 4.47 (septet,^3^*J*_H,H_ = 6.8 Hz, 1H, CH), 7.26 (s, 2H, aromatic), 7.36 (dt,^3^*J*_H,H_ = 7.1 Hz, ^4^*J*_H,H_ = 0.7 Hz, 2H, aromatic), 7.41–7.47 (m, 3H, aromatic), 7.80 (s, 1H, aromatic), 8.17 (d,^3^*J*_H,H_ = 7.8 Hz, 2H, aromatic), 8.50 (s, 2H, aromatic), 8.54 (s, 1H, aromatic), 8.77 (d,^3^*J*_H,H_ = 8.1 Hz, 1H, aromatic); ^13^C{^1^H}-NMR (CDCl_3_; 125.8 MHz) δ = 23.1, 23.3, 25.6, 27.9, 28.6, 109.6, 115.3, 115.6, 116.6, 120.6, 120.7, 121.1, 121.3, 122.2, 124.1, 126.5, 126.6, 129.4, 132.1, 132.4, 140.9, 143.6, 144.9, 148.49, 148.54, 149.4, 154.1, 156.4, 160.2, 160.6; MS (ES-TOF): *m*/*z* (rel. int.) M^+^ = 606.2502 (100%), (9-(8-isopropyl-2-methyl-5-nitrosoquinolin-6-yl)-9H-carbazole “VNS product” Scheme 3) Calcd for C_25_H_21_N_3_O (M − 2H)^+^ = 377.2294 (20%); HRMS (AP-TOF): *m*/*z* Calcd. for C_38_H_32_N_5_O_3_ M^+^ = 606.2505, Found 606.2505; UV-Vis (methanol; λ (nm) (logε)): 381 (4.46), 327 (4.17), 287 (4.57), 230 (4.86); IR (KBr): ν = 2962, 1600, 1529, 1452, 1316, 1228, 750 cm^−1^.

### 3.7. Syntheses of Aminoquinolines ***6***

Stannous chloride crystal (47.4 g, 250.0 mmol) was added to a stirred solution of nitroquinoline **4a**, **4b**, **4c,** or **4d** (25.0 mmol), respectively, and 6M hydrochloric acid (100 mL) in methanol (300 mL). After being stirred for 0.5 h at rt, the reaction mixture was heated to 40–50 °C for 2 h (only for **4b**) or was brought to reflux and stirred for 3 h. After cooling to rt, the mixture was basified with aqueous ammonia and extracted with chloroform (3 × 50 mL). The combined extract was dried over MgSO_4_ and evaporated to afford a solid (or liquid for **6e**), which was purified by crystallization from chloroform/hexane mixture to yield precipitates as follows (or was distilled bp 110–115 °C/ 3 mmHg for **6e** to afford red oil):

*Quinolin-8-amine* (**6a**) [23] as a beige solid 2.0 g (13.9 mmol, 57%); mp = 62–63 °C; ^1^H-NMR (CDCl_3_; 400.2 MHz) δ = 4.98 (bs, 2H, NH_2_), 6.93 (d, ^3^*J*_H,H_ = 7.5 Hz, 1H, aromatic), 7.15 (d, ^3^*J*_H,H_ = 8.1 Hz, 1H, aromatic), 7.29–7.39 (m, 2H, aromatic), 8.06 (d, ^3^*J*_H,H_ = 8.2 Hz, 1H, aromatic), 8.76 (d, ^3^*J*_H,H_ = 3.6 Hz, 1H, aromatic); ^13^C{^1^H}-NMR (CDCl_3_; 100.6 MHz) δ = 109.9, 115.9, 121.4, 127.2, 128.8, 135.9, 138.5, 143.9, 147.4.

*4-Chloroquinolin-8-amine* (**6b**) [24] as a yellow solid 3.3 g (18.8 mmol, 75%); mp = 90–95 °C; ^1^H-NMR (CDCl_3_; 400.2 MHz) δ = 4.94 (bs, 2H, NH_2_), 6.95 (d, ^3^*J*_H,H_ = 7.5 Hz, 1H, aromatic), 7.40 (t, ^3^*J*_H,H_ = 8.0 Hz, 1H, aromatic), 7.44 (d, ^3^*J*_H,H_ = 4.6 Hz, 1H, aromatic), 7.51 (dd, ^3^*J*_H,H_ = 8.4 Hz, ^4^*J*_H,H_ = 0.6 Hz, 1H, aromatic), 8.59 (d, ^3^*J*_H,H_ = 4.6 Hz, 1H, aromatic); ^13^C{^1^H}-NMR (CDCl_3_; 100.6 MHz) δ = 110.9, 112.1, 121.5, 127.2, 128.5, 139.0, 142.6, 144.3, 146.4; GC-MS: t_r_ = 6.3 min, (EI) M^+^ = 178 (100%); UV-Vis (methanol; λ (nm) (logε)): 363 (3.80), 340 (3.84), 291 (3.45), 253 (4.71), 207 (4.76); IR (KBr): ν = 3426, 3285, 1620, 1501, 1358, 808, 743 cm^−1^.

*8-(tert-Butyl)-2-methylquinolin-5-amine* (**6c**) as a brown solid, 2.7 g (12.8 mmol, 51%); mp._dec._ = 90.1–90.3 °C; ^1^H NMR (CDCl_3_; 500.2 MHz) δ = 1.63 (s, 9H, C(CH_3_)_3_), 2.70 (s, 3H, CH_3_), 3.96 (s, 2H, NH_2_), 6.65 (d,^3^*J*_H,H_ = 7.9 Hz, 1H, aromatic), 7.16 (d,^3^*J*_H,H_ = 8.6 Hz, 1H, aromatic), 7.40 (d,^3^*J*_H,H_ = 7.9 Hz, 1H, aromatic), 8.00 (d,^3^*J*_H,H_ = 8.6 Hz, 1H, aromatic); ^13^C{^1^H}-NMR(CDCl_3_; 125.8 MHz) δ = 25.4, 31.2, 35.9, 108.9, 117.9, 119.2, 126.1, 129.5, 138.4, 140.4, 147.4, 155.4; MS (ES-TOF): *m*/*z* (rel. int.) (M + H)^+^ = 215.1550 (100%); HRMS (ES-TOF): *m*/*z* Calcd for C_14_H_19_N_2_ (M + H)^+^ = 215.1548, Found 215.1550; UV-Vis (methanol; λ (nm) (logε)): 336 (3.82),252 (4.67), 204 (4.73); IR (KBr): ν = 3363, 2940, 1663, 1609, 1356, 823, 789 cm^−1^; Anal. Calcd for C_14_H_18_N_2_: C, 78.46; H, 8.47; N, 13.07; Found: C, 78.65; H, 8.49; N, 12.94.

*8-(iso-Propyl)-2-methylquinolin-5-amine* (**6d**) as a red liquid, 4.3 g (21.5 mmol, 86%); bp 110–115 °C/ 5 mm Hg; ^1^H-NMR (CDCl_3_; 400.2 MHz) δ = 1.32 (d, ^3^*J*_H,H_ = 7.0 Hz, 6H, CH(CH_3_)_2_), 2.69 (s, 3H, CH_3_), 3.92 (bs, 2H, NH_2_), 4.25 (septet,^3^*J*_H,H_ = 6.9 Hz, 1H, CH), 6.68 (d,^3^*J*_H,H_ = 7.8 Hz, 1H, aromatic), 7.13 (d,^3^*J*_H,H_ = 8.6 Hz, 1H, aromatic), 7.33 (d,^3^*J*_H,H_ = 7.8 Hz, 1H, aromatic), 7.96 (d,^3^*J*_H,H_ = 8.6 Hz, 1H, aromatic); ^13^C{^1^H}-NMR(CDCl_3_; 125.8 MHz) δ = 23.7, 25.5, 26.6, 109.5, 117.0, 119.9, 125.3, 129.7, 137.4, 139.8, 146.1, 157.2; MS (ES-TOF): *m*/*z* (rel. int.) (M + H)^+^ = 201.1393 (100%); HRMS (ES-TOF): *m*/*z* Calcd for C_13_H_17_N_2_ (M + H)^+^ = 201.1392, Found 201.1393; UV-Vis (methanol; λ (nm) (logε)): 340 (3.46), 253 (4.41), 202 (4.44); Anal. Calcd for C_13_H_16_N_2_: C, 77.96; H, 8.05; N, 13.99; Found: C, 77.60; H, 8.09; N, 13.86.

## 4. Conclusions

In this research, the synthesis of four new VNS substitution products and 8-nitroquinoline derivatives was presented. Two of these i.e., molecules **5a** and **5b,** illustrate the nitro/nitroso conversion within VNS substitution. Additionally, 9-(8-isopropyl-2-((8-isopropyl-2-methyl-5-nitroquinolin-6-yl)methyl)-5-nitrosoquinolin-6-yl)-9*H*-carbazole was presented as a double-VNS product. Our results showed that direct nucleophilic displacement of an aromatic hydrogen reaction proceeded together with expected S_N_Ar of chloride located in activated C4 position. Our findings show exclusively one product type, with a newly formed C7-N or C6-N bond in ortho-position to nitro group. Other regioisomers, especially in para-position, were not observed, which suggests the dominant role of the potassium cation in contributing attractive interaction to tight ion-pairs with the oxygen on nitro group. Additionally, a reduction of nitroquinoline derivatives by stannous chloride crystal as an effective and predictable reaction, tolerating other functional groups, was observed.

## Data Availability

Data set presented in this study is available in this article.

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
