# Peer review of "Direct Amination of Nitroquinoline Derivatives via Nucleophilic Displacement of Aromatic Hydrogen"

_molecules, 2021, doi:10.3390/molecules26071857_

Round 1

Reviewer 1 Report

The paper explored the VNS reaction on different nitro quinoline scaffolds. It is well written, even if there is a continuous shift between synthesis and XRD results, which maybe could be regroup in a single section of the paper. It is quite interesting how the managed to propose some explanation for the compounds obtained but here there are some remarks: the role of potassium, did the author thoughts to change potassium tert butoxide with some other base? Did the authors thought to use crown ether to make the potassium less available to the reaction medium? Secondly, it looks like the reaction mixture are quite complicated or the isolated yield quite low in some examples: there is presence of unreacted material in some cases? Or unidentifiable products?

On the intramolecular H bond proposed for 5b: I can even agree about its existence, based especially on the XRD, but in the NMR spectra the authors reported there is too much water (not a problem for the identity of the species) to explain that little chemical shifts difference among two solvents with an intramolecular hydrogen bond. If they want to study this aspect they should perform some IR studies, but I think it is just better to affirm this possibility through the XRD evidence.

On the absorption profile of 5b: the authors wrote about varying the pH, but in the end they showed an absorption spectrum with just KOH, maybe a full water pH titration should be interesting. The influence of the tautomeric equilibrium on the absorption profile is connected for different solvents to their more or less neutral nature (which is a strange statement), or to H bonding capability (better), but what about the different polarity/dielectric constant which is rather different for all these solvents?

I really liked the conclusion between 5b’ and 5b in the KOD/D2O 1H-NMR experiment.

Overall it is paper with a lot of ideas and information, but they could be better presented in term of manuscript layout content: maybe regroup synthesis, XRD and the rest in separate sections.

I suggest a revision especially directed to the presentation rather that to the scientific content which is interesting.

In supporting: what is the purpose of the NOESY of 4c which do not show any cross peak? It should be fixed and reported to show the spatial network of interaction of the proton atoms otherwise it should be removed. In addition they always show zoom of certain region in the various 3D spectra: the authors should show the overall 2D spectra and relative useful zoom of certain regions of interest. This apply to most of the 2D spectra reported in the supplementary materials.

Other minor things:

Line 34 “application's purpose” should be “the purpose of the application”

Line 52-54: I don’t think this link to previous work should be written here, maybe in the introduction fits better.

Line 59 “titled” should be “target”

Scheme 2 caption line 74: “(iii); H2SO4, HNO3 >70 °C” should be “…..T>70°C”

Scheme 1 is not a synthetic scheme but rather a Figure

I really think they should not omit all the synthetic details in Scheme 2 considering the purpose of the paper, but they should write all

Line 152: “To synthesis” should be “To synthesize”

Line 159: “was possibly” should be “was possible”

In Figure 3 (a) there is a full stop close to (a) and it should be removed

Line 319: there is a blue “an”

Remove appendix A and B from the MDPI template at the end

Author Response

A point-by-point response to comments from reviewers and editor

Reviewer #1

The paper explored the VNS reaction on different nitro quinoline scaffolds. It is well written, even if there is a continuous shift between synthesis and XRD results, which maybe could be regroup in a single section of the paper.

Response: We thank the reviewer for positive comments, constructive feedback, and recommendation for our work. The XRD results are grouped in two parts. One is dedicated to starting materials i.e., nitroquinoline derivatives, and the second to VNS type products. We intend to keep both groups of molecules separated. The VNS substitution presented by us is a valuable reaction with a rich potential future, so we would like to popularize it. We understand that the mechanism proposed by us is somewhat speculative at this stage of our research.

It is quite interesting how the managed to propose some explanation for the compounds obtained but here there are some remarks: the role of potassium, did the author thoughts to change potassium tert butoxide with some other base? Did the authors thought to use crown ether to make the potassium less available to the reaction medium?

Response: During amination reactions of chloro-phenanthrolines, we use NaH or tBuOK as a base. Our unpublished study shows the superiority of tBuOK. Thank you for the valuable comments; we have not performed experiments with systematic studies, including potassium, sodium, lithium, and their appropriate crown ethers. We do not possess the required crown ethers at the moment, and ordering them could take a somewhat long time. I will keep the idea in mind because it will be an excellent starting point for subsequent research. Takehiko Kawakami and Hitomi Suzuki (J. Chem. Soc., Perkin Trans. 1, 2000, 1259–1264) reported the reaction between 1,3-dinitrobenzene and 5-substituted derivatives with excess potassium or sodium methoxide in 1,3-dimethylimidazolidin-2-one (DMI). Their findings suggest that the reaction depends on the counter cation of methoxide anion. Moreover, the reaction does not take place with lithium methoxide as a base. Since 29/03/2020 (Monday), our university is closed due to Pandemia.

Secondly, it looks like the reaction mixture are quite complicated or the isolated yield quite low in some examples: there is presence of unreacted material in some cases? Or unidentifiable products?

Response: Yes, we always isolate unreacted material. The unreacted carbazole or phenothiazine makes isolation of products more difficult due to their compromised solubility. This is why we plan to use other amines for future studies dedicated to exploring the mechanism of VNS reaction. We want to use amines more suitable for NMR studies to exclude problems with the isolation of any possible unidentifiable or difficult to identify products.

On the intramolecular H bond proposed for 5b: I can even agree about its existence, based especially on the XRD, but in the NMR spectra the authors reported there is too much water (not a problem for the identity of the species) to explain that little chemical shifts difference among two solvents with an intramolecular hydrogen bond. If they want to study this aspect they should perform some IR studies, but I think it is just better to affirm this possibility through the XRD evidence.

Response: We thank the reviewer for constructive ideas. We have already performed some IR studies. However, the results are not satisfactory yet. We are planning to perform some NMR studies consisting of deuterium incorporation into the product. The presence of deuterium in the "hydrogen bond" should reduce its strength. We believed that it would be excellent evidence for the presented results. I have attached the requested IR spectra, please see the attachment.

On the absorption profile of 5b: the authors wrote about varying the pH, but in the end they showed an absorption spectrum with just KOH, maybe a full water pH titration should be interesting. The influence of the tautomeric equilibrium on the absorption profile is connected for different solvents to their more or less neutral nature (which is a strange statement), or to H bonding capability (better), but what about the different polarity/dielectric constant which is rather different for all these solvents?

Response: Yes, we check the possibility of using water for pH titration. However, molecule 5b was insoluble. We add the data for THF, and we add dielectric constant data.

I really liked the conclusion between 5b’ and 5b in the KOD/D21H-NMR experiment.

Response: We thank the reviewer for the positive comments. We appreciate it.

Overall it is paper with a lot of ideas and information, but they could be better presented in term of manuscript layout content: maybe regroup synthesis, XRD and the rest in separate sections.

Response: We thank the reviewer for positive comments, constructive feedback, and recommendation for our work. The XRD results are grouped in two parts. One is dedicated to starting materials i.e., nitroquinoline derivatives, and the second to VNS type products. We intend to do not mix both groups of molecules.

I suggest a revision especially directed to the presentation rather that to the scientific content which is interesting.

In supporting: what is the purpose of the NOESY of 4c which do not show any cross peak? It should be fixed and reported to show the spatial network of interaction of the proton atoms otherwise it should be removed. In addition they always show zoom of certain region in the various 3D spectra: the authors should show the overall 2D spectra and relative useful zoom of certain regions of interest. This apply to most of the 2D spectra reported in the supplementary materials.

Response: We modify SM part.

Other minor things:

Line 34 “application's purpose” should be “the purpose of the application”

Response: The suggested correction has been made. Thank you so much for your help. We appreciate it.

Line 52-54: I don’t think this link to previous work should be written here, maybe in the introduction fits better.

Response: The suggested correction has been made. Thank you so much for your help. We appreciate it.

Line 59 “titled” should be “target”

Response: The suggested correction has been made. Thank you so much for your help. We appreciate it.

Scheme 2 caption line 74: “(iii); H2SO4, HNO3 >70 °C” should be “…..T>70°C”

Response: The suggested correction has been made. Thank you so much for your help. We appreciate it.

Scheme 1 is not a synthetic scheme but rather a Figure

Response: The suggested correction has been made. Thank you so much for your help. We appreciate it.

I really think they should not omit all the synthetic details in Scheme 2 considering the purpose of the paper, but they should write all

Response: The suggested correction has been made. Thank you so much for your help. We appreciate it.

Line 152: “To synthesis” should be “To synthesize”

Response: The suggested correction has been made. Thank you so much for your help. We appreciate it.

Line 159: “was possibly” should be “was possible”

Response: The suggested correction has been made. Thank you so much for your help. We appreciate it.

In Figure 3 (a) there is a full stop close to (a) and it should be removed

Response: The suggested correction has been made. Thank you so much for your help. We appreciate it.

Line 319: there is a blue “an”

Response: The suggested correction has been made. Thank you so much for your help. We appreciate it.

Remove appendix A and B from the MDPI template at the end

Response: The suggested correction has been made. Thank you so much for your help. We appreciate it.

Some improvements in the writing have been made. We have revised the whole manuscript carefully and tried to avoid any grammar or syntax errors. Besides, we have asked several skilled authors of English language papers to check the English. Thank you so much for your help. We appreciate it

Yours sincerely, Jacek Nycz (on behalf of all co-authors)

Reviewer 2 Report

Jacek E. Nycz and the co-authors described a nucleophilic aromatic substitution (VNS) reaction in electron-deficient nitroquinolines. This study show that direct nucleophilic displacement reaction of an aromatic hydrogen proceeds faster than expected SNAr of chloride located in activated C4 position. The results are interesting and the products are correctly described and analyzed. Although it is very targeted it can probably be extended to other derivatives and can be of use to the scientific community. This is the reason why I recommend the publication of this article in molecules. 

Author Response

Dear Madam or Sir,
On behalf of all authors, I want to thank the referee for the manuscript's approval and the valuable comments about the suitability of its publication.
With kind regards,
Jacek Nycz

Reviewer 3 Report

The paper is devoted to amination of nitroquinolines by nucleophilic aromatic substitution reactions with amines.

The experimental procedures are described clearly, supporting information contains all details and complete spectral data.

At the same time, there are some critical points that should not be ignored:

lines 141-143: “Our results showed that the direct nucleophilic displacement of an aromatic hydrogen reaction proceeds faster than expected the SNAr of chloride located in activated C4 position.”

-this important conclusion is not substantiated by any experimental data, presented in paper. No kinetic data, no intermediate detected, products were not isolated, only detected in complex reaction mixture.

-lines 114-115: “we isolated crystalline 9-(8-nitroquinolin-7-yl)-9H-carbazole (5a) and (Z)-7-(9H-carbazol-9-yl)-8-(hydroxyimino)quinolin-5(8H)-one (5b) with moderate yield”  The yields, presented in Scheme 3 are not “moderate”, but rather low (5% and 16%).

Scheme 3 is poorly justified by experimental data. It is not clear how A is transformed into 5b’. OH group in 5-position can result, in principle, from nucleophilic substitution with hydroxide ion. It is not clear from the scheme whether presented yield of 5b is from or 4b. The possibility for transformation 4b-à 5b can be readily proved experimentally.  From the scheme presented one can conclude that sigma complex B is formed and remains unchanged. Is it true? In the text authors stated that the attack of carbazole anion occurs only at the 7-position.

Also, some inaccuracies were identified in the manuscript:

line 13: nucleophilic aromatic substitution (VNS)  - in abstract

lines 39-40: nucleophilic aromatic substitution (NASH) - in introduction, i.e different abbreviations for the same process.

-Scheme 1. Caption to the picture is somewhat misleading.

Presented in Scheme are not “structures” but rather “structural formula”, and they can be readily derived from the presented names of compounds A and B. It would be more appropriate to present reaction scheme, by which they were obtained. Otherwise it can be removed altogether.

  • Scheme 2. There is no “R” at the left side (comps 1, 4a,b)! What means R near arrows?

Section 2.4. What means “…directly  accepts hydride ion aromatization of σH-adduct”? nitro group can accept protone rather than hydride ion.

lines 103-104: “ipso replacement of nitro substituent can occur and may compete with SNAr substitution”
-substitution of nitro substituent with nucleophile is itself SNAr process, therefore it may not compete with itself,  but rather with nucleophilic aromatic substitution of hydrogen (VNS).

Some identified inaccuracies and/or the places needed corrections, are highlighted in yellow in the attached file.

In general, it seems to me that the main value of paper is the preparation of several substituted derivatives of quinoline and their extensive structural and spectral characterization (x-ray structure, stacking, electronic and NMR spectra.). Reasoning about the mechanism are poorly justified experimentally and are rather speculative.

As the mechanism of presented reactions was not studied in details, I think that discussion of mechanism in the manuscript should be reduced.

I think that the paper may be published in Molecules after consideration the above points.

Author Response

A point-by-point response to comments from reviewers and editor

Reviewer #3

The paper is devoted to amination of nitroquinolines by nucleophilic aromatic substitution reactions with amines.

The experimental procedures are described clearly, supporting information contains all details and complete spectral data.

Response: We thank the reviewer for positive comments, constructive feedback, and recommendation for our work.

At the same time, there are some critical points that should not be ignored:

lines 141-143: “Our results showed that the direct nucleophilic displacement of an aromatic hydrogen reaction proceeds faster than expected the SNAr of chloride located in activated C4 position.”

this important conclusion is not substantiated by any experimental data, presented in paper. No kinetic data, no intermediate detected, products were not isolated, only detected in complex reaction mixture.

Dear Reviewer, thank you for the valuable comments. We have not performed sufficient experiments to study VNS and SNAr reaction's speed systematically; they could take a somewhat long time. I will keep the idea in mind because it will be an excellent starting point for subsequent research. According to Mąkosza et al., the VNS often proceeds much faster than the aromatic nucleophilic substitution of halogen SNAr, as mentioned in the References part [Acc. Chem. Res. 1987, 20, 282–289, Chem. Commun., 1999, 689–690, Tetrahedron 1998, 54, 8797–8810]. However, in literature, we can find some exceptions. One example could be 4-fluoro-5-nitropyridine which participates exclusively in the aromatic nucleophilic substitution of halogen SNAr [J. Med. Chem. 2018, 61, 9371–9385]. Since 29/03/2020 (Monday), our university is closed due to Pandemia.

lines 114-115: “we isolated crystalline 9-(8-nitroquinolin-7-yl)-9H-carbazole (5a) and (Z)-7-(9H-carbazol-9-yl)-8-(hydroxyimino)quinolin-5(8H)-one (5b) with moderate yield”  The yields, presented in Scheme 3 are not “moderate”, but rather low (5% and 16%).

Response: The suggested correction has been made. Thank you so much for your help. We appreciate it.

Scheme 3 is poorly justified by experimental data. It is not clear how A is transformed into 5b’. OH group in 5-position can result, in principle, from nucleophilic substitution with hydroxide ion. It is not clear from the scheme whether presented yield of 5b is from or 4b. The possibility for transformation 4b-à 5b can be readily proved experimentally.  From the scheme presented one can conclude that sigma complex B is formed and remains unchanged. Is it true? In the text authors stated that the attack of carbazole anion occurs only at the 7-position.

Response: The suggested correction has been made. Thank you so much for your help. We appreciate it. According to literature data, adduct B could be transformed into adduct A, and vice versa. Unfortunately, we do not identify the second possibly regioisomers i.e. 9-(8-nitroquinolin-5-yl)-9H-carbazole.

Also, some inaccuracies were identified in the manuscript:

line 13: nucleophilic aromatic substitution (VNS)  - in abstract

Response: The suggested correction has been made. Thank you so much for your help. We appreciate it.

lines 39-40: nucleophilic aromatic substitution (NASH) - in introduction, i.e different abbreviations for the same process.

Response: The suggested correction has been made. Thank you so much for your help. We appreciate it.

Scheme 1. Caption to the picture is somewhat misleading.

Presented in Scheme are not “structures” but rather “structural formula”, and they can be readily derived from the presented names of compounds A and B. It would be more appropriate to present reaction scheme, by which they were obtained. Otherwise it can be removed altogether.

Response: The suggested correction has been made. Thank you so much for your help. We appreciate it.

Scheme 2. There is no “R” at the left side (comps 1, 4a,b)! What means R near arrows?

Response: The suggested correction has been made. Thank you so much for your help. We appreciate it.

Section 2.4. What means “…directly  accepts hydride ion aromatization of σH-adduct”? nitro group can accept protone rather than hydride ion.

Response: The suggested correction has been made. Thank you so much for your help. We appreciate it.

lines 103-104: “ipso replacement of nitro substituent can occur and may compete with SNAr substitution”
-substitution of nitro substituent with nucleophile is itself SNAr process, therefore it may not compete with itself,  but rather with nucleophilic aromatic substitution of hydrogen (VNS).

Response: The suggested correction has been made. Thank you so much for your help. We appreciate it.

Some identified inaccuracies and/or the places needed corrections, are highlighted in yellow in the attached file.

Response: The suggested correction has been made. Thank you so much for your help. We appreciate it.

In general, it seems to me that the main value of paper is the preparation of several substituted derivatives of quinoline and their extensive structural and spectral characterization (x-ray structure, stacking, electronic and NMR spectra.). Reasoning about the mechanism are poorly justified experimentally and are rather speculative.

As the mechanism of presented reactions was not studied in details, I think that discussion of mechanism in the manuscript should be reduced.

I think that the paper may be published in Molecules after consideration the above points.

Response: The suggested correction has been made. Thank you so much for your help. We appreciate it.

Some improvements in the writing have been made. We have revised the whole manuscript carefully and tried to avoid any grammar or syntax errors. Besides, we have asked several skilled authors of English language papers to check the English. Thank you so much for your help. We appreciate it

Yours sincerely, Jacek Nycz (on behalf of all co-authors)

Reviewer 4 Report

The authors report some novel heterocyclic motives obtained via nucleophilic aromatic substitution of Nitroquinoline derivatives. In general the synthesis of novel heterocycles is a worthwhile goal and the compounds presented herein are interesting, also considering the reported properties. Also the molecules are properly characterized so overall I think the paper may fit within the scope of the journal.

However I first of all found it really hard to follow the story of the paper and i.e. the introduction can be improved by making a more illustrative Scheme 1 (showing really the underlying chemistry and not just the targets) as well as providing a more clear illustration about the goals of the work. Otherwise it was hard to identify what was actually planned. So maybe some careful rewriting of the introduction and goals section would be beneficial.

Also I do not really understand about the meaning of section 2.1. ?! At least after reading it several times it does not make any meaning to me.

Then in terms of presentation I recommend to present the schemes with more details..e.g. as already mentioned - Scheme 1 should indicate how these compounds were formed and not just the targets -  Scheme 2 should give the reagents and sequences in more detail and so on - Scheme 3 and 4 _ structures 4 and carbazole should be given to clarify, ...

Then I wonder why only carbazole was used as a nucleophile. In my opinion some other amine nucleophiles should be tested either.

The influence of the counter cation K+ should be proven by using other bases with smaller cations! Maybe the regioselectivity can be improved thereby.

The reaction with 4-Cl-derivatives should be properly discussed in a scheme in the main manuscript and in my opinion it is not valid  to argue yet that the substitution of H over Cl is so much favoured because of the inherently higher reactivity of the 5 and 7 position - maybe the authors can use 5- or 7-Cl and test this one to see if the selectivity can be improved.

So overall it is valuable research but I recommend some detailed revision.

Author Response

A point-by-point response to comments from reviewers and editor

Reviewer #4

The authors report some novel heterocyclic motives obtained via nucleophilic aromatic substitution of Nitroquinoline derivatives. In general the synthesis of novel heterocycles is a worthwhile goal and the compounds presented herein are interesting, also considering the reported properties. Also the molecules are properly characterized so overall I think the paper may fit within the scope of the journal.

Response: We thank the reviewer for positive comments, constructive feedback, and recommendation for our work.

However I first of all found it really hard to follow the story of the paper and i.e. the introduction can be improved by making a more illustrative Scheme 1 (showing really the underlying chemistry and not just the targets) as well as providing a more clear illustration about the goals of the work. Otherwise it was hard to identify what was actually planned. So maybe some careful rewriting of the introduction and goals section would be beneficial.

Response: The suggested correction has been made. Thank you so much for your help. We appreciate it.

Also I do not really understand about the meaning of section 2.1. ?! At least after reading it several times it does not make any meaning to me.

Response: The suggested correction has been made. Thank you so much for your help. We appreciate it.

Then in terms of presentation I recommend to present the schemes with more details..e.g. as already mentioned - Scheme 1 should indicate how these compounds were formed and not just the targets -  Scheme 2 should give the reagents and sequences in more detail and so on - Scheme 3 and 4 _ structures 4 and carbazole should be given to clarify, ...

Response: The suggested correction has been made. Thank you so much for your help. We appreciate it.

Then I wonder why only carbazole was used as a nucleophile. In my opinion some other amine nucleophiles should be tested either.

Response: Yes, we plan to use other amines than carbazole or phenothiazine for future studies dedicated to exploring the mechanism of VNS reaction. We want to use other amines than carbazole more suitable for NMR studies to exclude problems with the isolation of any possible unidentifiable products. At the moment, we do not have the required reagents and the ordered of them, and carry out reactions could take a somewhat long time, but I will keep the idea in mind because it will be an excellent starting point for another subsequent paper. Since 29/03/2020 (Monday), our university is closed due to Pandemia.

The influence of the counter cation K+ should be proven by using other bases with smaller cations! Maybe the regioselectivity can be improved thereby.

Response: During conducting reaction aminations of chloro-phenanthrolines, we use NaH or tBuOK, as a base. Our not yet presented study show superior of tBuOK. Thanks for the valuable comments; we have not performed experiments with systematic studies, including potassium, sodium, lithium, and their appropriate crown ethers. Takehiko Kawakami and Hitomi Suzuki (J. Chem. Soc., Perkin Trans. 1, 2000, 1259–1264) reported the reaction between 1,3-dinitrobenzene and 5-substituted derivatives with excess potassium, sodium, and lithium methoxide in 1,3-dimethylimidazolidin-2-one (DMI). They found that the counter cation of methoxide depends on and the lithium salt proved ineffective. The reaction does not take place with lithium methoxide as a base.

The reaction with 4-Cl-derivatives should be properly discussed in a scheme in the main manuscript and in my opinion it is not valid  to argue yet that the substitution of H over Cl is so much favoured because of the inherently higher reactivity of the 5 and 7 position - maybe the authors can use 5- or 7-Cl and test this one to see if the selectivity can be improved.

Response: Thanks for the valuable comments, we will perform experiments to systematic studies that includes the effect of. Dear Reviewer, we have not performed sufficient experiments to study VNS and SNAr reaction speed systematically; they could take a somewhat long time. I will keep the idea in mind because it will be an excellent starting point for subsequent research. According to Mąkosza et al., the VNS often proceeds much faster than the aromatic nucleophilic substitution of halogen SNAr, as mentioned in the References part [Acc. Chem. Res. 1987, 20, 282–289, Chem. Commun., 1999, 689–690, Tetrahedron 1998, 54, 8797–8810]. However, in literature, we can find some exceptions. One example could be 4-fluoro-5-nitropyridine which participates exclusively in the aromatic nucleophilic substitution of halogen SNAr [J. Med. Chem. 2018, 61, 9371–9385].

So overall it is valuable research but I recommend some detailed revision.

Response: We thank the reviewer for positive comments, constructive feedback, and recommendation for our work. Thank you so much for your help. We appreciate it.

Some improvements about the writing have been done. We have revised the whole manuscript carefully and tried to avoid any grammar or syntax error. In addition, we have asked several colleagues who are skilled authors of English language papers to check the English. Thank you so much for your help. We appreciate it.

Yours sincerely, Jacek Nycz (on behalf of all co-authors)

Round 2

Reviewer 1 Report

Thanks for the detailed reply and for the work done in the revision process. Solid work.

Reviewer 4 Report

Revision was carried out nicely and questions answered convincingly so I think the paper is suited for publication now.